# Cellular and Humoral Immune Profiles After Hepatitis E Vaccination and Infection

**DOI:** 10.3390/v17070901

**Published:** 2025-06-26

**Authors:** Joakim Øverbø, Jennifer L. Dembinski, Toril Ranneberg Nilsen, Vethanayaki Sriranganathan, Veselka Petrova Dimova-Svetoslavova, Asma Aziz, K Zaman, Cathinka Halle Julin, Firdausi Qadri, Kathrine Stene-Johansen, Taufiqur Rahman Bhuiyan, Warda Haque, Susanne Dudman

**Affiliations:** 1Norwegian Institute of Public Health, NO-0213 Oslo, Norwaykathrine.stene-johansen@fhi.no (K.S.-J.); 2Department of Microbiology, Institute of Clinical Medicine, University of Oslo, NO-0424 Oslo, Norway; 3Oslo University Hospital, NO-0424 Oslo, Norwayvesdim@ous-hf.no (V.P.D.-S.); 4International Centre for Diarrheal Diseases Research (ICDDR,B), Dhaka 1212, Bangladesh; kzaman@icddrb.org (K.Z.);; 5International Vaccine Institute, Seoul 08826, Republic of Korea

**Keywords:** hepatitis E virus, HEV, vaccine, immunology, IgG, T-cell

## Abstract

Hepatitis E virus (HEV) causes significant morbidity and mortality globally, particularly affecting vulnerable populations such as pregnant women. HEV239 (Hecolin^®^), a recombinant vaccine containing the immunodominant protruding (E2) domain of the HEV capsid protein, has demonstrated effectiveness, yet detailed human cellular immune responses remain understudied. This study characterized humoral and cellular immune responses following vaccination with HEV239 or natural HEV infection in healthy Bangladeshi women aged 16–39 years. Using dual IFNγ and IL-4 ELISpot assays, we found robust, predominantly Th1-mediated cellular responses at 30 days after the third vaccine dose, comparable to responses during acute infection. Longitudinal antibody assessments confirmed sustained antibody production, primarily against the E2 domain of genotypes 1 and 3, persisting up to two years post-vaccination. Despite limitations related to sample size and assay sensitivity, our findings underscore the immunogenic potential of HEV239 and support a broader use in HEV-endemic regions.

## 1. Introduction

Hepatitis E virus (HEV), the causative agent of hepatitis E infection, is a quasi-enveloped, single-stranded RNA virus comprising three or four open reading frames (ORFs) [1]. These ORFs encode nonstructural proteins (ORF1), the capsid protein (ORF2), and a phosphoprotein (ORF3) that facilitates viral release from host cells. The ORF2 protein includes three linear structural domains: the shell (S or N), middle (M), and protruding (P, E2 or C) domains. The C domain is regarded as the immunodominant epitope of the capsid [2].

A truncated recombinant fragment of the HEV genotype 1 ORF2 protein (aa 368–606), designated p239, can be expressed in Escherichia coli, where it self-assembles into 23 nm virus-like particles that form the antigenic component of the HEV239 vaccine (Hecolin^®^, Xiamen Innovax Biotech Co., Ltd., Xiamen, China) [3,4]. The Wantai HEV IgG ELISA employed in this study uses the same region to capture anti-HEV IgG, allowing sensitive, cross-genotype antibody detection [5].

Within the genus *Paslahepevirus*, eight genotypes (HEV-1–HEV-8) are recognized [6]. These genotypes are serologically cross-reactive due to shared immunodominant do-mains, likely enabling cross-genotype protection by vaccines developed from a single genotype [1]. HEV-1 and HEV-2 are restricted to humans and spread primarily by fecally contaminated water, causing large outbreaks and severe acute hepatitis, with case-fatality rates of up to 25% in pregnant women [1,7]. HEV-3 and HEV-4 are zoonotic; humans acquire infection sporadically through consumption of undercooked pork, game, or via direct contact with infected animals. Illness is often subclinical but can progress to chronic hepatitis and cirrhosis in immunosuppressed individuals [1]. HEV-5 and HEV-6 circulate in wild boar, while HEV-7 and HEV-8 infect dromedary and Bactrian camels; of these, only HEV-7 has been documented to cause hepatitis in humans to date [8].

Despite substantial evidence supporting the crucial role of cell-mediated immunity (CMI) in viral clearance across various viral infections [9,10], very limited data exist regarding CMI in the context of HEV infection [11,12]. Production of T-helper type 1 (Th1) cytokines, such as interferon gamma (IFNγ), is typically associated with viral clearance or resistance, whereas Th2 cytokines, including interleukin-4 (IL-4), have been linked to disease progression [13]. Both CD4+ and CD8+ T-cell responses targeting HEV ORFs are elicited during natural infection. Furthermore, the important role of T-cell responses in eliminating an infection is apparent in the development of chronic infections among people receiving T-cell depleting immunosuppressants [14,15].

Studies investigating CMI following natural HEV infection have shown Th1 responses preferentially targeting the ORF2 protein, particularly evidenced by increased interferon-γ (IFNγ) production [15]. While animal model studies have demonstrated HEV239 vaccine-induced IFNγ-mediated cellular responses [16], human data regarding cellular immune responses after vaccination remain sparse [17].

HEV239 has been found to elicit a strong and long-lasting antibody response, persisting for over eight years in most recipients [18]. The relationship between antibody production and cellular response is, however, unclear. A small study of four participants receiving HEV239 found a specific response against the E2 antigenic site and cross-reaction with other genotypes [19], but if this applies in a wider population and over time, including previously infected people is uncertain.

A pilot study conducted in rural Bangladesh demonstrated that a two-dose regimen of the HEV239 vaccine was highly immunogenic, inducing strong HEV-specific antibody responses that persisted for at least two years [17]. The study also detected HEV-specific T-cell responses 30 days after vaccination. Building upon these initial findings, a large-scale Phase 4, double-blind, cluster-randomized trial was conducted in rural Bangladesh to assess the safety and effectiveness of the HEV239 vaccine among women of childbearing age. This study demonstrated that the vaccine was safe and effective in preventing hepatitis E infection in this population and confirmed the vaccine’s ability to elicit a strong antibody response [20].

The current study, conducted on a subgroup of participants from this Phase 4 trial, aims to provide a detailed characterization of the cellular and humoral immune responses following HEV239 vaccination and natural HEV infection. Using dual IFNγ and IL-4 ELISpot assays and longitudinal antibody measurements, we seek to better understand the magnitude, persistence, and coordination of vaccine-induced immunity in humans.

## 2. Materials and Methods

### 2.1. Study Population

This study included 50 healthy women aged 16 to 39 years, randomly selected from a larger cohort of 19,460 women participating in the main HEV 239 vaccine trial conducted in Matlab, Bangladesh. The parent trial included women from 67 villages randomized to receive either the HEV239 vaccine (Hecolin^®^, Innovax, China) or a control hepatitis B vaccine (HBV) (HEPA-B, Incepta, Dhaka, Bangladesh) according to a three-dose schedule administered at Day 0, Day 30, and Day 180 [19]. Additionally, six participants from the control group with confirmed HEV infection were included.

Ethical approval was obtained from the icddr,b Research Review Committee (RRC) and Ethical Review Committee (ERC), the Directorate General of Drug Administration (DGDA) in Bangladesh, and the Regional Ethics Committee (REC) in Norway. All participants provided written informed consent prior to enrollment. Detailed methodological details can be found in the study protocol [21] and the Phase 4 article [20].

### 2.2. Sample Collection and Processing

Venous blood samples (up to 9 mL) were collected from each participant at three time points: baseline (Day 0, prior to vaccination), Day 210 (30 days after the third vaccine dose), and Day 730 (two years after the last vaccination).

Peripheral blood mononuclear cells (PBMCs) and plasma were isolated within hours of collection via Ficoll-Paque density gradient centrifugation (Ficoll-Paque Premium 1.077; GE Healthcare, Chicago, IL, USA), using SepMate tubes (Stemcell Technologies, Vancouver, BC, Canada) according to the manufacturer’s instructions. Isolated PBMCs were counted and cryopreserved in aliquots of ≤10 million cells/vial at −150 °C using a freezing medium composed of 25% fetal calf serum, 10% dimethyl sulfoxide (DMSO), and 65% AIM-V medium (Cat No 12055-091, GIBCO, Grand Island, NY, USA). Plasma samples were stored at −80 °C until further use. Samples were transported internationally under strictly controlled cold-chain conditions using a liquid nitrogen dry shipper (−196 °C).

### 2.3. Thawing Procedure, Viability, and Recovery

On the day of assay, PBMCs were thawed rapidly in a 37 °C water bath and immediately washed twice with warm (37 °C) AIM-V medium (Cat No 12055-091, GIBCO). Viability and cell counts were determined using an automated cell counter (Countess 3 Automated Cell Counter, Invitrogen, Carlsbad, CA, USA). Cell samples with viability ≥ 65% were selected for ELISpot assays, ensuring that each assay contained a minimum of 375,000 live cells per well. PBMCs were plated in duplicate whenever cell numbers permitted, but when fewer than 750,000 viable cells were available for a given time point, only a single replicate well was set up for that sample.

### 2.4. ELISpot Assay (IFNγ/IL-4)

Study-specific standard operating procedures (SOPs) for the collection and analysis of PBMCs were developed based on the manufacturer’s recommendations and previously published protocols [10]. Cell viability was assessed using Trypan Blue exclusion prior to plating. HEV-specific IFNγ and IL-4 T-cell responses were measured by an ex vivo ELISpot assay according to the manufacturer’s instructions (Human IFNγ/IL-4 Double-Color ELISPOT, ImmunoSpot, Cleveland, OH, USA, Mabtech, Stockholm, Sweden). A commercially available peptide pool consisting of 163 overlapping peptides (15 mers with 11 aa overlap) comprising the entire HEV ORF2 region (5 µg/mL; PepMix™ HEV (ORF2), JPT Peptide Technologies, Berlin, Germany) was used for antigen-specific T-cell stimulation.

In short, 375,000 PBMC in AIM-V medium (Cat No 12055-091, GIBCO) were added to each well in 96-well plates precoated with IFNγ and IL-4 antibodies. Antigenic stimulants used were in duplicate as follows: negative control (0.25% DMSO in AIM-V to match the final DMSO concentration of the JPT peptide pool), positive control (100 ng/mL anti-CD3, Mabtech, Sweden), and the JPT peptide pool (PepMix™ HEV (ORF2) JPT Peptide Technologies, Germany) at 5 µg/mL in DMSO/AIM-V. Plates were incubated for 48 h at 37 °C in a humidified incubator with 5% CO2 then developed. The plates were read using a CTL S6 UltraV ImmunoSpot analyzer (Cellular Technology Limited, Shaker Heights, OH, USA). Antigen-specific responses were calculated by subtracting the mean number of spots in the negative control wells from the test (JPT) wells and the results were expressed as spot-forming units (SFU)/10^6^ PBMCs. A positive ELISpot reaction was defined as >2 SD over the average number of spot-forming units (SFU) in the negative control wells in accordance with published recommendations [22]. Results below this number were assigned a value of 0.014. Participant samples with no response to the positive control were excluded.

### 2.5. Serological Analyses

HEV-specific IgG antibody levels were measured using the Wantai HEV IgG ELISA kit (Beijing Wantai Biological Pharmacy, Beijing, China) following the manufacturer’s instructions. To minimize assay variability, paired samples from the same individual were analyzed within a single run. Each run included serial dilutions of the WHO reference reagent for HEV antibodies (WHO 95/584), allowing conversion of optical density (OD) values into WHO units per milliliter (WU/mL) using a five-parameter logistic function [23]. Samples were classified as positive if OD values exceeded the established cut-off (OD/CO > 1; 0.2 WU/mL). The lower limit of quantification (LLOQ) was defined at OD/CO = 0.03 (0.06 WU/mL); values below LLOQ were assigned a value of 0.014. Samples exceeding the assay range (OD/CO = 25) were retested following dilution.

Detailed antibody characterization was further performed using the RecomLine HEV IgG immunoblot assay (Mikrogen Diagnostik, Neuried, Germany), designed to qualitatively and semi-quantitatively detect HEV-specific IgG antibodies against recombinant antigens representing two major HEV genotypes (HEV-1, HEV-3). Each nitro-cellulose strip carries two separate recombinant proteins for every antigenic domain, one from genotype 1 and one from genotype 3. The following four antigenic regions are included: O2N (N-terminal part of ORF2), O2M (Middle region of ORF2), O2C (C-terminal part of ORF2, including E2, the basis for the HEV 239 vaccine), O3 (ORF3 protein, HEV-1 and HEV-3). Assays were performed according to manufacturer guidelines, with automated and visual interpretation of bands based on the manufacturer’s scoring system. Results are given in OD values over cut-off, and results below cut-off were assigned a value of 0.014.

### 2.6. Statistical Analysis

Geometric mean titers (GMTs) with 95% confidence intervals (CIs) were calculated for Wantai antibody measurements and cellular responses following log-transformation. Mean values with 95% CIs were used for Recomline results. Continuous variables were assessed using a non-parametric (Mann–Whitney U test) method. Spearman rank correlation was employed to evaluate associations between antibody and cellular responses. Statistical significance was defined as *p* < 0.05. Data analysis was performed using STATA (version 18).

## 3. Results

Out of 50 participants randomly selected for the immunological substudy, 49 donated blood at baseline (Day 0), with subsequent sample availability varying by time point and sample type (Figure 1). The following results describe the humoral and cellular immune responses observed across these time points.

At baseline (Day 0), 19 of the 49 participants (38.8%, 95% CI: 26.0–53.3%) were seropositive for HEV antibodies. Of these, 9 were in the HEV239 group (36.0%, 95% CI: 19.6–56.5%) and 10 in the HBV-vaccinated control group (41.6%, 95% CI: 23.7–62.2%). All participants in the HEV239 group were seropositive on Day 210 while three participants became seronegative two years later (Day 730).

We observed a significant increase in the geometric mean titers (GMT) of HEV-specific IgG antibodies in the HEV239 group one month after the third vaccine dose. The GMT increased from a baseline level of 0.1 WU/mL (95% CI: 0.04–0.22) to 33.39 WU/mL (95% CI: 22.0–50.7) on Day 210, while remaining stable in the control group (Figure 2).

PBMC samples were obtained at baseline (Day 0) and 210 days after initial vaccination (Day 210). Valid T-cell response data from at least one of these time points were available for 23 participants (see Figure 1). At Day 210, IFNγ-producing T-cell responses were significantly elevated in HEV239 recipients compared to the HBV-vaccinated control group (GMT 8.4, 95% CI: 0.43–163.9 vs. 0.01, 95% CI: 0.01–0.01 SFU/10^6^ PBMC; *p* = 0.007). Although an increase from baseline (GMT 0.60, 95% CI: 0.00–789.4 SFU/10^6^ PBMC) was detected, this was not statistically significant. IL-4 responses were also increased but lower (non-significant), indicating a predominantly Th1-mediated immune profile (see Figure 2).

PBMCs from three individuals with acute HEV infection demonstrated robust IFNγ responses (GMT 56.5, 95% CI: 2.1–1495.3 SFU/10^6^ PBMC) accompanied by lower IL-4 responses, indicating a predominantly Th1-mediated cellular response similar to that seen in vaccinated participants.

Spearman correlation analyses revealed a positive association between antibody titers (WU/mL) and T-cell responses, Spearman rank correlation was rs = 0.602 (*p* = 0.001) for IFNγ (left) and rs = 0.470 (*p* = 0.013) for IL-4 (right), indicating a moderate, significant positive correlation between humoral and cellular responses. Several IgG positive samples had an undetectable T-cell response (see Figure 3).

Longitudinal antibody responses measured using both Wantai (WU/mL) and RecomLine assays showed clear increases from Day 0 to Day 210 in the HEV239 group, followed by a decline at Day 730. Nevertheless, antibody levels at Day 730 remained elevated above baseline, indicating sustained immunogenicity in most vaccinated individuals (Figure 4). In the HEV239 group, the geometric mean concentration of IgG measured by Wantai increased from 0.09 WU/mL (95% CI: 0.04–0.22) at Day 0 to 33.39 WU/mL (95% CI: 21.98–50.72) at Day 210, and remained at 1.26 WU/mL (95% CI: 0.67–2.36) at Day 730. The control group showed no meaningful change over time, with geometric means remaining below 0.10 WU/mL at all time points.

Antibody responses measured by RecomLine were restricted to the C-terminal part (O2C) of the ORF2 protein, with clear increases observed only in the HEV239 group. Geometric mean OD/CO values for genotype 3 increased from 0.033 (95% CI: 0.014–0.076) at Day 0 to 8.96 (95% CI: 7.91–10.15) at Day 210 and declined to 0.96 (95% CI: 0.45–2.08) at Day 730. The response to genotype 1 followed a similar but less pronounced pattern, with geometric means of 0.026 (95% CI: 0.014–0.048), 5.43 (95% CI: 4.31–6.84), and 0.19 (95% CI: 0.07–0.51) at the respective time points. No significant increase in antibodies was observed in the control group for either genotype (see Figure 4).

Antibodies directed against other ORF2 regions (N-terminal (N) and middle (M)) and ORF3 were negative in all participants except one (for both genotype 1 and 3). This individual was an HBV-vaccinated participant who showed low-level ORF3 (genotype 1) reactivity at baseline (OD/CO 1.6) and Day 210 (OD/CO 1.3); the signal had waned below cut-off by Day 730 and is consistent with previous natural HEV exposure

Two participants demonstrated a baseline cellular response to HEV, although due to unavailability of a functional follow-up sample, a potential booster effect from vaccination could not be confirmed. These individuals’ serological responses aligned with the observed cellular reactivity.

Two participants provided valid antibody and cellular responses on both Day 0 and 210. Both received the HEV vaccine and had a strong response on all measurements except against other parts of ORF 2 or 3 than O2C (see Table 1).

## 4. Discussion

Our findings demonstrate that vaccination with HEV239 (Hecolin^®^) elicits robust and coordinated humoral and cellular immune responses comparable to those induced by natural HEV infection. Specifically, strong IFNγ-producing Th1 cellular responses were identified following vaccination, aligning well with responses observed in individuals recovering from acute HEV infection.

Previous studies have documented increased T-cell responses targeting the ORF2 capsid protein in individuals recovering from HEV infection, highlighting the immunogenic importance of this antigen [15,24]. In this context, our results align with earlier research indicating that the capsid protein, especially its protruding C domain, contains major immunodominant epitopes capable of inducing strong immune responses. Our use of a comprehensive peptide pool spanning the entire ORF2 region confirmed the capacity of the vaccine to stimulate antigen-specific IFNγ production significantly. The lower IL-4 responses compared to IFNγ further confirm a predominantly Th1-skewed immune response induced by vaccination, consistent with the profile observed during self-limiting natural infection and previous HEV239 findings [17,25].

Correlation analysis revealed significant, yet moderate, associations between antibody titers and T-cell responses. Notably, discrepancies were observed where some participants had high antibody levels but low or absent cellular responses, and vice versa. These discrepancies might reflect true biological variability, such as T-cell-independent antibody production, or could result from assay sensitivity limitations. Future studies with larger sample sizes and more sensitive cellular assays will be needed to clarify the underlying mechanisms.

Our longitudinal antibody assessments demonstrated sustained antibody responses up to two years post-vaccination, reinforcing the durability and long-term potential of vaccine-induced immunity. The RecomLine assays consistently detected sustained antibody levels against both genotype 1 and 3, but almost exclusively against O2C (E2). This demonstrates the specificity of HEV239 in inducing antibodies against its antigen without boosting antibodies against other parts of HEV, and also the immune dominance of this antigen in natural HEV infection.

The robust antibody response we observed against the O2C domain of both genotype 1 and 3 suggests that HEV239 can induce meaningful cross-genotype immunity. Nonetheless, even within the immunodominant E2s (p239) region, genotypes 1–4 share only 86–92% amino-acid identity (≈14% divergence) [26], so complete interchangeability cannot be assumed. This nuance is borne out experimentally: the genotype-1-based HEV239 fully protected rhesus macaques and participants in the large Chinese phase III trial against genotype 4 challenge or infection [4,27], yet pig studies showed only partial protection against genotype 3 unless a matched p239-Genotype 3 construct was used [28]. Hence, while broad cross-genotype protection is achievable, vaccine performance may vary with circulating strains. Clinical studies in genotype 3-dominant settings and evaluation of multivalent or genotype-adapted p239 formulations will be crucial to ensure uniformly high effectiveness.

The recombinant hepatitis B vaccine contains only hepatitis B surface antigen (HBsAg), a protein that shares negligible amino-acid identity with the HEV ORF2 capsid proteins and for which no cross-reactive T- or B-cell epitopes with HEV have been reported. Accordingly, we did not expect, and did not observe, any HEV-specific cellular or humoral boosting in the HBV-vaccinated control group, confirming its suitability as a negative control.

Several limitations should be acknowledged. First, the effective sample size for cellular assays was reduced as many PBMC specimens had too low cell yield, poor viability, or lack of response to the positive control, limiting statistical power. Variability in sample handling, cryopreservation, and thawing could have affected PBMC functionality and contributed to the quality problems [29]. Second, cellular immunity was assessed only up to Day 210, precluding evaluation of longer-term T-cell durability beyond six months. Third, the study population comprised healthy women aged 16–39 years from a single rural setting, so findings may not generalize to other demographics or regions.

Despite these limitations, our findings support the capacity of HEV239 vaccination to induce robust and long-lasting Th1-skewed cellular and humoral immune responses comparable to natural HEV infection. These results encourage broader implementation and further investigation of HEV239 vaccination in HEV-endemic areas.

## Figures and Tables

**Figure 1 viruses-17-00901-f001:**
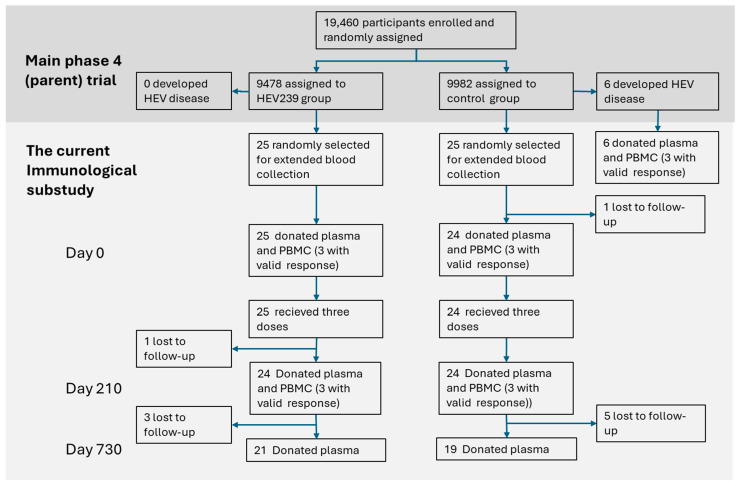
Flowchart showing participant selection and availability of plasma and viable PBMC samples at baseline (Day 0), Day 210, and Day 730 in the immunological substudy of the phase 4 HEV vaccine trial. Samples from six participants with acute HEV infection are also included.

**Figure 2 viruses-17-00901-f002:**
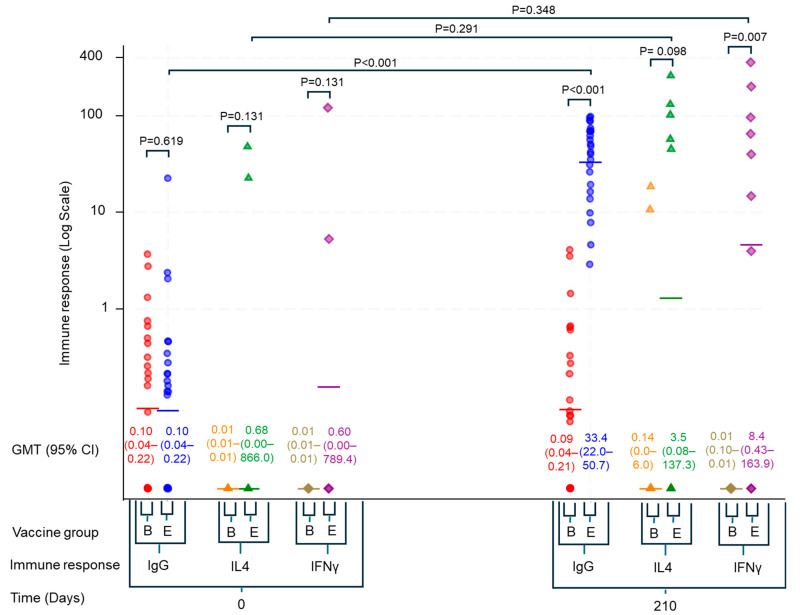
Immune responses at baseline (Day 0) and 30 days after the third vaccine dose (Day 210) in HEV239 vaccine recipients (E) and HBV-vaccinated controls (B). IgG antibody levels (circles) are expressed in WU/mL; IL-4 (triangles) and IFNγ (diamonds) T-cell responses are expressed as spot-forming units (SFU) per 10^6^ PBMCs. Data are shown on a log scale with geometric mean titers (GMT) and 95% confidence intervals. *p*-values compare responses between and within groups.

**Figure 3 viruses-17-00901-f003:**
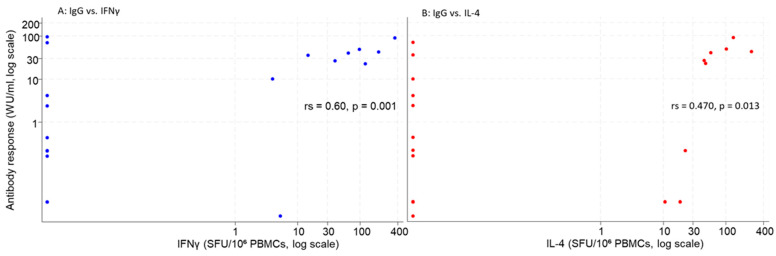
Correlation between antibody response (IgG, WU/mL) and T-cell responses (IFNγ and IL-4, SFU/10^6^ PBMCs) in vaccine recipients. Both axes are presented on a logarithmic scale. Spearman rank correlation was rs = 0.602 (*p* = 0.001) for IFNγ (**left**) and rs = 0.470 (*p* = 0.013) for IL-4 (**right**), indicating a moderate, significant positive correlation between humoral and cellular responses.

**Figure 4 viruses-17-00901-f004:**
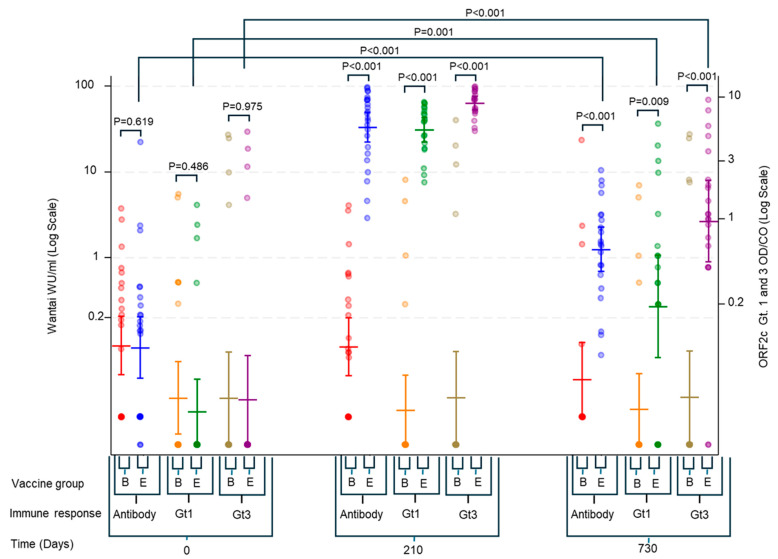
IgG antibody responses measured by Wantai ELISA (WU/mL, left y-axis) and RecomLine immunoblot (OD/CO, right y-axis) targeting the C-terminal ORF2 region (O2C) of HEV genotypes 1 (Gt1) and 3 (Gt3) at baseline (Day 0), 30 days after the third vaccine dose (Day 210), and two years after initial vaccination (Day 730). Data are shown for HEV-vaccinated participants (E) and HBV controls (B). Horizontal bars indicate geometric means with 95% CI. Statistical comparisons between and within groups are shown above each time point.

**Table 1 viruses-17-00901-t001:** Individual-level humoral and cellular immune responses at baseline (Day 0) and post-vaccination (Day 210) in two participants who received the HEV239 vaccine and provided valid PBMC samples at both time points. Values are reported for anti-HEV IgG (WU/mL), antibody reactivity to ORF2c genotype 1 and 3 (OD/CO), and cytokine-producing T-cells (IL-4 and IFNγ) measured as spot-forming units (SFU) per 10^6^ PBMCs.

Participant	Vaccine	Immune Response	Day 0	Day 210
1	HEV	WU/mL	2.40	48.66
		ORF2c gt.1	0.7	9.2
		ORF2c gt.3	2.7	12.2
		IL4	0.014	102.67
		IFNγ	0.014	97.34
2	HEV	WU/mL	0.19	9.97
		ORF2c gt.1	0.014	4.2
		ORF2c gt.3	0.014	8.0
		IL4	0.014	22.66
		IFNγ	0.014	4.0

## Data Availability

The data presented in this study are available on request from the corresponding author due to privacy issues.

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
