# Peer review of "Cellular and Humoral Immune Profiles After Hepatitis E Vaccination and Infection"

_viruses, 2025, doi:10.3390/v17070901_

Round 1
Reviewer 1 Report
Comments and Suggestions for Authors
The paper represents a multidisciplinary work assessing cellular immunity after Hepatitis E (HEV) immunization of a small group of Bangladesh females with a recombinant antigen-based Hecolin® vaccine. T-cell-mediated cellular response (in vitro IFNγ and IL-4 production) was evaluated with a standard ELISpot assay. The authors have revealed post-vaccination Th1-mediated cellular reaction in most cases, as well as long-term antibody response which covered, mostly, viral genotypes 1 and 3.
Remarks:
Under Materials and Methods, one should mention if this multi-step approach to blood MNC isolation/cryopreservation/ELISpot assay is standard and reproducible, may be, referring to some existing laboratory guidelines.
Line 120: The stain used for cell viability assay should be mentioned.
Line 121: “cell samples” would be better than “cells”.
Line 123: “a single replicate per condition” means absence of parallel assays in some cases?
Line 140: One should note if the cutoff values in ELISpot results were assessed by appropriate standards (e.g., from the reagent manufacturer)
Line 210: the group of HBV-vaccinated controls should be also described in more details since HBV vaccination may boost both cellular and humoral immune response.
Line 221: Please explain the near-zero values of IFNγ and IL-4 production in sufficient part of vaccinated persons in Fig.3 (due to problems with assay technique, or absence of cell immune response?)
Some moderate copy editing is necessary, to make the content more clear and understandable.
Comments on the Quality of English LanguageModerate copy editing is required
Reviewer 2 Report
Comments and Suggestions for Authors
Hepatitis E virus (HEV) is a major cause of acute viral hepatitis, primarily transmitted through the fecal-oral route. Effective prevention strategies are essential to lower the occurrence of HEV globally. Although a recombinant HEV vaccine exists, it is currently licensed and available only in China and Pakistan. More research is needed to assess the need for broader vaccination.
The study of Øverbø et al. investigates the immune responses elicited by the HEV239 (Hecolin®) vaccine. The focus is on its effectiveness in generating both humoral (antibody-mediated) and cellular immune responses among Bangladeshi women, a group at higher risk for severe HEV-related illness. The study is extremely relevant and contributes to a more complete survey of the HEV vaccine. However, there are some gaps in the study's presentation that need to be addressed.
L32 HEV is a quasi-enveloped (eHEV) virus. Please clear the information
L36 m- M
L40, Please, mention the expression system of p239
L41 The sentence is unclear; please explain the WantaiHEV IgG Assay in more detail.
L43 The explanation of HEV classification is not complete. Please describe the different genotypes, their transmission methods, and the severity of the development of a HEV infection
L46 reference is missing
L 52 reference is missing
L81 reference is missing
L97 icddr,b ?
L192 – Please insert Figure 2 immediately after its citation
L242 I do not understand the sentence. Please provide more data.
How do you separately measure the antigenic response to the HEV-1 and HEV-3, as a RecomLine Kit is loaded with both Ag (HEV-1 and HEV-3)?
Figure 4 is not cited in the text
Please include in the discussion a possible concern regarding the Hecolin vaccine and the antigenic variability of the ORF2 protein across genotypes. Although the HEV genotypes are generally considered to belong to a single serotype, suggesting that cross-protection against different HEV strains should be achievable, this protection may not be uniform.
Please give examples of cross-protective immunity across different mammalian HEV genotypes.
Round 2
Reviewer 2 Report
Comments and Suggestions for Authors
I am satisfied with the authors’ responses and revisions. I recommend the manuscript for publication.